# In Vivo Efficacy of SQ109 against *Leishmania donovani*, *Trypanosoma* spp. and *Toxoplasma gondii* and In Vitro Activity of SQ109 Metabolites

**DOI:** 10.3390/biomedicines10030670

**Published:** 2022-03-14

**Authors:** Kyung-Hwa Baek, Trong-Nhat Phan, Satish R. Malwal, Hyeryon Lee, Zhu-Hong Li, Silvia N. J. Moreno, Eric Oldfield, Joo Hwan No

**Affiliations:** 1Host-Parasite Research Laboratory, Institut Pasteur Korea, Seongnam-si 13488, Korea; kyunghwa.baek@ip-korea.org (K.-H.B.); trongnhat.phan@ip-korea.org (T.-N.P.); hyeryon.lee@ip-korea.org (H.L.); 2Department of Chemistry, University of Illinois at Urbana-Champaign, Urbana, IL 61801, USA; satishm@illinois.edu (S.R.M.); eoldfiel@illinois.edu (E.O.); 3Center for Tropical and Emerging Global Diseases, Department of Cellular Biology, University of Georgia, Athens, GA 30602, USA; zhli@uga.edu (Z.-H.L.); smoreno@uga.edu (S.N.J.M.)

**Keywords:** SQ109, in vivo, *Trypanosoma cruzi*, *Trypanosoma brucei*, *Leishmania donovani*, metabolites

## Abstract

SQ109 is an anti-tubercular drug candidate that has completed Phase IIb/III clinical trials for tuberculosis and has also been shown to exhibit potent in vitro efficacy against protozoan parasites including *Leishmania* and *Trypanosoma cruzi* spp. However, its in vivo efficacy against protozoa has not been reported. Here, we evaluated the activity of SQ109 in mouse models of *Leishmania, Trypanosoma* spp. as well as *Toxoplasma* infection. In the *T. cruzi* mouse model, 80% of SQ109-treated mice survived at 40 days post-infection. Even though SQ109 did not cure all mice, these results are of interest since they provide a basis for future testing of combination therapies with the azole posaconazole, which acts synergistically with SQ109 in vitro. We also found that SQ109 inhibited the growth of *Toxoplasma gondii* in vitro with an IC_50_ of 1.82 µM and there was an 80% survival in mice treated with SQ109, whereas all untreated animals died 10 days post-infection. Results with *Trypanosoma brucei* and *Leishmania donovani* infected mice were not promising with only moderate efficacy. Since SQ109 is known to be extensively metabolized in animals, we investigated the activity in vitro of SQ109 metabolites. Among 16 metabolites, six mono-oxygenated forms were found active across the tested protozoan parasites, and there was a ~6× average decrease in activity of the metabolites as compared to SQ109 which is smaller than the ~25× found with mycobacteria.

## 1. Introduction

Diseases caused by kinetoplastid parasite infections, such as leishmaniasis, Chagas disease, and human African trypanosomiasis (HAT), are neglected tropical diseases for which there are urgent demands to deliver safer and more effective therapies [1]. One potential path to develop new therapies is to repurpose existing drugs or drug leads, used to treat other diseases, to the disease of interest. For example, amphotericin B (Figure 1), used to treat fungal infections, and miltefosine, originally developed as an anti-cancer drug, have been successfully repurposed for treating visceral leishmaniasis (VL) [1,2]. Another example is fexinidazole, very recently approved (https://www.fda.gov/media/72973/download, accessed on 6 January 2022) to treat HAT caused by *Trypanosoma brucei gambiense*, which has also recently completed a Phase II proof-of-concept study for treating Chagas disease [3,4]. There are also other examples of anti-parasitic drug candidates at much earlier stages of development. For example, the anti-antiarrhythmic drug amiodarone has both in vitro and in vivo activity against *T. cruzi* and acts synergistically with posaconazole, in vitro, leading to improved activity in vivo, in mice [5]. In more recent work, the combination therapy of amiodarone with itraconazole was found to give very promising results in a study involving >100 dogs [6]. However, amiodarone can have adverse side-effects, so there is interest in investigating other drugs/drug candidates that in protozoa have similar mechanisms of action to amiodarone: protonophore uncoupling, Ca^2+^ homeostasis and in some cases, effects on sterol composition [7].

One such example is the anti-tubercular drug candidate SQ109 [*N*-adamantan-2-yl-*N*’-((E)-3,7-dimethyl-octa-2,6-dienyl)-ethane-1,2-diamine] which has been found have potent activity against kinetoplastid parasites in vitro [7,8,9,10,11], but there have been no reports of its activity in vivo in any protozoa. In *Mycobacterium tuberculosis*, SQ109 inhibits cell wall biosynthesis by targeting the trehalose monomycolate transporter MmpL3, Mycobacterial protein Large 3, in addition to acting as a protonophore uncoupler [12]. In kinetoplastid parasites, SQ109 was found to be active against *T. brucei* (half-maximal inhibitory concentration, IC_50_ = 0.078 µM, in promastigotes); *T. cruzi* (IC_50_ = 50 nM in trypomastigotes; 4.6 µM in epimastigotes, and ~0.5 to 1 µM in intracellular amastigotes); *L. mexicana* (IC_50_ = ~11 nM in intracellular amastigotes and ~500 nM in promastigotes), and *L. donovani* (IC_50_ = 7.17 nM in intracellular amastigotes and 630 nM in promastigotes), in vitro [8,9,10,11]. In these organisms, one of the suggested mechanisms is again that the compound acts as a protonophore uncoupler and rapidly collapses the mitochondrial membrane potential [9,10,11]. In addition, as with the kinetoplastid parasites, SQ109 was also found to have potent in vitro activity against *Plasmodium falciparum* where it was shown to selectively inhibit mature gametocytes (stage IV/V), the stage responsible for transmission, with an IC_50_ value of 0.105 µM [13]. With the generation of an SQ109-resistant clone, mutations in *Pf*VapA (a V-type H+-ATPase) were identified and suggested as a potential drug target for further validation [13]. SQ109 growth inhibition has not been explored in other apicomplexans, such as *T. gondii*, an important protozoan parasite that causes toxoplasmosis in humans. SQ109 thus appears to have potential to be repurposed for treating parasitic diseases, however, experiments to date have been limited to in vitro studies. In this report, we thus evaluate the in vivo efficacy of SQ109 in treating infections caused by kinetoplastid parasites as well as by *T. gondii*, in order to examine its potential as an anti-parasitic agent. Furthermore, we investigated the in vitro activity of 16 likely metabolites of SQ109 detected experimentally, to evaluate whether the metabolites might contribute to the activity of SQ109 against pathogenic protozoa,

## 2. Materials and Methods

### 2.1. Compounds

For *T. brucei* assays, pentamidine (Sigma) was dissolved in dimethyl sulfoxide (DMSO) and used as a reference drug [14]. For *T. cruzi* assays, benznidazole (Biosynth Carbosynth, Newbury, UK) was prepared in DMSO and used as a reference drug [15]. For *Leishmania donovani* assays, miltefosine (MedChemExpress, Monmouth Junction, NJ, USA) was prepared in water and used as a reference drug [16]. The final concentrations of DMSO never exceeded 0.6% and 10% in in vitro and in vivo assays, respectively, and was therefore not toxic to the parasite, mammalian cells, or mice.

### 2.2. In Vitro Activity Assay for T. brucei

The bloodstream form (BSF) of *T. b. brucei* Lister strain 427 was cultivated in HMI-9 medium supplemented with 10% heat-inactivated fetal bovine serum (FBS, Gibco, Waltham, MA, USA), 100 µg/mL penicillin, and 100 µg/mL streptomycin (1% P/S, Gibco) at 37 °C and a 5% CO_2_ atmosphere. Parasites were sub-cultured every 3 or 4 days and maintained for 10 passages. Growth inhibition of *T. b. brucei* was assayed by measuring the conversion of resazurin to resorufin. The assays were performed in 384-well plates that were seeded with *T. brucei* 427 BSF (5 × 10^4^ cells per well). After seeding, the parasites were exposed to the compounds for 3 days. Then, 200 µM of resazurin sodium salt (Sigma) was added, and the samples were incubated for 5 h. After incubation, the parasites were fixed using 4% paraformaldehyde, and the plates were analyzed using a Spectramax plate reader at 590 nm (emission) and 530 nm (excitation) [17].

### 2.3. In Vitro Activity Assay for T. cruzi

#### 2.3.1. Tissue Culture Trypomastigotes Assay

LLC-MK2 monkey kidney cells and U2OS human osteosarcoma cells were maintained in Dulbecco’s modified Eagle’s medium (DMEM, Welgene, Gyeongsangbuk-do, Korea) medium supplemented with high glucose, 10% FBS, and 1% penicillin/streptomycin (P/S) at 37 °C with 5% CO_2_ in air [18]. LLC-MK2 for the tissue-culture derived trypomastigotes (TCTs) assay was maintained in DMEM with low glucose, 2% FBS, and 1% P/S, at 37 °C with 5% CO_2_ in air and used as host cells for amplification of Dm28c strain. LLC-MK2 cells were infected with frozen stock of TCTs of the Dm28c strain in low glucose DMEM supplemented with 2% FBS and 1% P/S at 37 °C with 5% CO_2_ in air [19]. TCTs were obtained from the supernatant 6~7 days after infection by centrifugation at 2000× *g* for 10 min. To test activity of drugs, trypomastigotes were treated with different concentrations of drugs or 0.1% DMSO as a negative control and incubated for 24 h at 37 °C with 5% CO_2_ in air. The viable parasites were counted in a Neubauer chamber or assayed by resazurin as described in the *T. brucei* assay.

#### 2.3.2. Epimastigotes Assay

*T. cruzi* epimastigotes were cultured in liver infusion tryptose medium (BD Difco™, FisherScientific Ida, Porto Salto, Portugal) with 10% FBS at 28 °C. The exponential growth phase of epimastigotes were treated with different concentrations of drugs or 0.1% DMSO as negative control and incubated for 3 days at 28 °C. The viability of parasites was quantified with resazurin as described in the *T. brucei* assay.

#### 2.3.3. Intracellular Amastigotes Assay

U2OS human osteosarcoma cells were maintained in DMEM supplemented with high glucose, 10% FBS, and 1% P/S at 37 °C with 5% CO_2_ in air. U2OS cells were infected with TCTs at a multiplicity of infection (MOI) of 1:4 and treated with drugs simultaneously in 384-well plates. After 3 days of drug treatment, cells were then fixed with 4% paraformaldehyde (PFA), washed with 1× PBS, and stained with a fluorescent probe (Draq-5, Thermo Fisher, Waltham, MA, USA). Cell images were acquired by an automated image analyzer (Operetta, Perkin Elmer Technology, Waltham, MA, USA); analysis included at least 1000 cells in five fields per well. For the imaging of infected cells and parasites, Draq-5 was observed under a 20× air objective. An image analysis algorithm (Columbus, Perkin Elmer Technology) was used to detect the Draq-5 signal in the nuclei of cells and parasites as described in a previous article and shown in Appendix A [20]. The number of parasites was defined by the value of the number of parasites in infected cells of the acquired image.

### 2.4. In Vitro Activity Assays for L. donovani

*L. donovani* strain Ld1S (MHOM/SD/62/1S-CL2D) parasites were maintained in Syrian Golden hamsters. Anesthetized, 5-week-old hamsters were inoculated with 10^8^ metacyclic promastigotes of *L. donovani* by intracardiac injection. The infected hamsters were euthanized using CO_2_ when their weight decreased by 15% to 20%. Their spleens were collected and homogenized in cold PBS, and tissue debris was removed by centrifugation at 130× *g* for 5 min at 4 °C. The parasites were then harvested by centrifugation of the supernatant at 2000× *g* for 10 min at 4 °C. After three washes with PBS, the parasites were further isolated by Percoll gradient [21]. After centrifugation at 3500× *g* for 45 min at 15 °C, parasites were collected from the interface of the gradient, then washed three times with PBS. These purified amastigotes were used to infect THP-1 or differentiated into promastigotes.

#### 2.4.1. Promastigotes Assay

Promastigotes were maintained at 28 °C in modified M199 culture medium (Sigma, St. Louis, MO, USA) with 20 mM HEPES (Gibcco, Waltham, MA, USA), 0.1 mM adenine (Sigma), 0.0005% hemin (Sigma), 0.0001% biotin (Sigma), 0.0002% biopterin (Santa Cruz, Dallas, TX, USA), and 4.62 mM NaHCO_3_ (Sigma), supplemented with 10% FBS and 1% P/S. The cultures were diluted every 7–10 days and underwent no more than five passages to avoid generation of genetic variability [22]. *L. donovani* promastigotes were treated with different concentrations of drugs or 0.1% DMSO as negative control and incubated for 3 days at 28 °C. The viability of parasites was quantified with resazurin as described for the *T. brucei* assay.

#### 2.4.2. Intracellular Amastigotes Assay

The THP-1 human monocyte cell line was grown in RPMI-1640 medium (Welgene, Gyeongsangbuk-do, Republic of Korea) supplemented with 10% FBS and 1% P/S. For differentiation, THP-1 cells were treated for 3 days with 50 ng/mL of PMA [20]. After being plated in 384-well plates, THP-1 cells were infected with amastigotes or stationary phase promastigotes at MOI of 1:10 and treated with drugs after 1 day of infection. After 3 days of drug treatment, cells were then fixed with 4% PFA, washed with 1× PBS and stained with Draq-5. An image analysis was performed as described in the *T. cruzi* assay (Appendix A).

### 2.5. In Vitro Activity Assay for T. gondii

We used a parasite strain expressing red fluorescent protein (td-tomato) for in vitro drug testing [23,24]. Tachyzoites were maintained in human fibroblasts (hTert cells). Human fibroblasts were cultured in 96-well plates for 24 h prior to the addition of 4000 fluorescent tachyzoites/well. Fluorescence values were measured for 3 to 4 days, and both excitation (544 nm) and emission (590 nm) were read from the bottom of the plates in a Molecular Devices (Molecular Devices, LLC, CA, USA) plate reader.

### 2.6. T. brucei in Vivo Model and Drug Efficacy Test

Five-week-old female BALB/c mice were infected with *T. b. brucei* Lister 427 (4 × 10^4^ cells) by i.p. injection. The mice were divided into groups (*n* = 5), and drug treatment was performed for five consecutive days by starting from day 1 post-infection and administering 30 mg/kg of pentamidine, or 30 mg/kg of SQ109, or 100 mg/kg of SQ109, respectively. Drugs were freshly prepared each day. All drugs were administered once daily for 5 days via the per os (p.o.) route. Parasitemia was evaluated daily for 2 weeks by blood collection from the mouse tail vein, and survival was monitored for 1 month. Mice showing impaired health status and/or with a parasite load of >10^8^ cells per mL of blood were euthanized.

### 2.7. T. cruzi In Vivo Model and Drug Efficacy Test

BALB/c mice were infected with TCT of *T. cruzi* Dm28c (10^7^ cells) by i.p. injection. Five days post-infection, infected mice were sacrificed to collect BSF of *T. cruzi*. BALB/c mice were intraperitoneally infected with BSF of *T. cruzi* Dm28c (5 × 10^4^ cells). Five days post-infection, BALB/c mice were infected with previously collected 2nd-round BSF cells (3 × 10^4^ cells). Mice were divided into groups (*n* = 10) and treated orally with drugs for 5 consecutive days for 2 weeks by starting from day 2 post-infection administration of DPBS, 100 mg/kg of benznidazole, or 100 mg/kg of SQ109 via *p.o.* route, respectively. Parasitemia was evaluated every 2 days for 5 weeks by blood collection from the mouse tail vein, and survival was monitored for 2 months. Mice showing impaired health status were euthanized.

### 2.8. L. donovani In Vivo Model and Drug Efficacy Test

BALB/c mice (*n* = 5) were injected with 2 × 10^7^ hamster spleen-derived *L. donovani* amastigotes via the retro-orbital venous sinus route. From day 7 post-infection, groups of mice were treated using the drug vehicle only, miltefosine (30 mg/kg), or SQ109 (30, 50, 100 mg/kg). On day 16 post-infection, all animals were humanely euthanized and assessed microscopically using Giemsa-stained liver imprints. Parasite burdens were measured (blinded to treatment) by counting the number of amastigotes per 1000 cell nuclei and multiplying this number by the liver weight (mg) (LDU) [25]. The LDU values for the drug-treated samples were compared to those of the untreated samples.

### 2.9. T. gondii In Vivo Model and Drug Efficacy Test

Experiments were carried out using 20 fresh *T. gondii* tachyzoites of the RH strain to infect Webster mice. Drugs were dissolved in 10% Kolliphor HS 15 and were inoculated intraperitoneally. Treatment was initiated 6 h after infection and administered daily for 10 days with 20 and 50 mg/kg of SQ109 (i.p. or p.o.). Surviving mice were challenged with 1000 RH tachyzoites 42 days after infection. Survival was monitored for 2 months.

### 2.10. Statistical Analysis

The values produced by the image algorithm were further analyzed using GraphPad Prism 6 (GraphPad Software) for graphical representations and half-maximal effective concentration (EC_50_) value determinations.

### 2.11. Ethics

Female BALB/c mice (5–6 weeks old; body weight 15–20 g) and male Golden Syrian hamsters (5–6 weeks old; body weight 60–80 g) were purchased from Orientbio Inc. (Seongnam-si, Korea) and Central Laboratory Animal Inc. (Seoul, Korea), respectively. All animal handling and experiments were performed in compliance with the guidelines and principles established by the Korean Animal Protection Law (https://elaw.klri.re.kr/eng_service/lawTwoView.do?hseq=32497, accessed on 7 March 2022). All protocols for animal experiments were reviewed and approved by the Institutional Animal Care and Use Committee (IACUC, protocol #IPK-19002 for acute *T. brucei*, #IPK-17006-2 for *T. cruzi,* and #IPK-16003-3 for VL) of the Institut Pasteur Korea. Mouse experiments in Georgia followed a reviewed and approved protocol by the Institutional Animal Care and Use Committee (IACUC) and followed U.S. Government principles for the Utilization and Care of Vertebrate Animals, Animal Protocol A2018 02-021.

## 3. Results

### 3.1. In Vitro Activity of SQ109 against T. brucei, T. cruzi, and L. donovani

The structures of SQ109 and the reference compounds used in this study are shown in Figure 1 and, for reference, their in vitro activity against the *T. brucei*, *T. cruzi*, and *L. donovani* strains used in the in vivo studies is shown in Figure 2. The growth inhibition of *T. brucei* was assessed in trypomastigote form by measuring viability. With 3 days of compound incubation, the IC_50_ values of SQ109 and pentamidine were 0.320 and 0.150 µM, respectively, Figure 2A, comparable to that in a previous report [8]. For *T. cruzi*, the activity of SQ109 was assessed for trypomastigotes, epimastigotes, and intracellular amastigotes, with benznidazole as a control. For trypomastigotes, IC_50_ values of benznidazole and SQ109 were 59.7 and 2.65 µM, respectively (Figure 2B). In the epimastigote assay, the growth inhibition was observed by resazurin reduction, and the IC_50_ values were 12.6 µM for SQ109 and 3.25 µM for benznidazole (Figure 2C). For the amastigote assay, human osteosarcoma U2OS cells infected with the metacyclic trypomastigotes were used. After 3 days of compound incubation, the images of stained cells were acquired and analyzed for the quantification of intracellular amastigote and host cell number (see Appendix A). The IC_50_ value of SQ109 against intracellular *T. cruzi* survival was 0.610 µM and the half-maximal cytotoxic concentration (CC_50_) value against the host U2OS was 2.95 µM. The benznidazole control showed an IC_50_ value of 1.71 µM, so SQ109 was more potent (Figure 2D). Viscerotropic species of *L. donovani* were used for the anti-leishmanial in vitro activity assessment. The IC_50_ values of SQ109 and the control compound, miltefosine, against promastigote growth were 0.175 µM and 0.171 µM, respectively (Figure 2E).

For the intracellular amastigotes assay, differentiated THP-1 (a monocytic leukemia cell line) was infected with *L. donovani* amastigotes followed by 3 days of compound incubation. Based on the quantified stained cell images, the IC_50_ value of SQ109 against the amastigote was 1.38 µM, and no host cell toxicity was observed up to 10 µM (Figure 2F and Appendix A). The IC_50_ value for miltefosine was 1.51 µM, and no host cell toxicity was observed at the highest tested dose (Figure 2F).

### 3.2. In Vivo Activity of SQ109 against T. brucei

After confirming the activity of SQ109 in these in vitro tests, we then tested SQ109 against a kinetoplastid parasite, *T. brucei*, in an in vivo model. For the *T. brucei* acute model, BALB/c mice were infected with the bloodstream form of the parasite, and the parasitemia in blood as well as the survival of the mice were determined. In the SQ109 treated group (30 mg/kg, quaque die for 5 days, per os administration), the average day of survival was 8.6, which is 3.4 days of extended survival compared to the vehicle control (Figure 3A). Increasing the dose of SQ109 to 100 mg/kg showed a similar (3.6) average days of extended survival (Figure 3A). However, even though SQ109 successfully suppressed the parasite load in the blood for several days, parasitemia rapidly increased after the termination of SQ109 administration (Figure 3B).

### 3.3. In Vivo Activity of SQ109 against T. cruzi

BALB/c mice infected with the bloodstream form of *T. cruzi* were used for the *T. cruzi* acute model. As in the *T. brucei* study, parasitemia and survival were evaluated. The peak of parasitemia for the vehicle-treated group was seen at 14 days post-infection, and 46.1% of parasite reduction was seen with the group treated with 100 mg/kg of SQ109 via oral gavage (q.d. for 2 weeks). The peak of parasitemia for the drug-treated group was 18 days post-infection, which is 4 days of delay compared to that of the vehicle control (Figure 4A). The benznidazole-treated group showed complete suppression of parasitemia during the entire experimental period. In terms of survival, 80% of SQ109-treated mice survived, whereas the vehicle group showed only 30% survival. Benznidazole protected all the mice from death at 40 days after infection (Figure 4B).

### 3.4. In Vivo Activity of SQ109 against L. donovani

For the VL acute mouse model, *L. donovani* amastigotes isolated from infected hamster spleens were used to infect BALB/c mice by the retro-orbital route, and the drugs were administered for 5 days starting from 5 days post-infection. After 5 days of drug treatment, mice were sacrificed for the assessment of Leishman Donovan Units (LDUs) from collected livers. The parasite burden (in LDU) of the vehicle-treated group was 1132 ± 73, and those of SQ109-treated mice were 1060 ± 110, 772 ± 89, and 737 ± 145, for the 30-, 50-, and 100-mg/kg dosed groups, respectively. Even though there was a significant reduction of parasite burden with 50- and 100-mg/kg SQ109 dosings (31.8% and 34.9%, respectively), SQ109 was not an efficacious compared when compared with the miltefosine treatment, which showed an LDU of 142 ± 24 (87.5% parasite reduction) (Figure 5A). As seen in the Giemsa-stained liver impression smears, amastigotes are still seen in the SQ109-treated mice, whereas a dramatic decrease of parasites is observed in the miltefosine-treated group (Figure 5B).

### 3.5. In Vitro and In Vivo Activity of SQ109 against T. gondii

As shown in Figure 6A, we found an IC_50_ of 1.82 μM in *T. gondii* in vitro assay. In vivo efficacy results are shown in Figure 6B,C. All untreated mice died 10 days post-infection, but there was ~50–80% survival of the i.p. SQ109-treated mice (Figure 6B) and 30–40% survival at 40 days post-infection using oral gavage (Figure 6C). Thus, SQ109 kills *T. gondii* in vitro and has promising activity in vivo. At day 42, we challenged the live animals with 100 *T. gondii* parasites, a control to verify that they were infected initially. The animals (4 out of 6) treated with 20 mg/kg/day survived the challenge because the lower dose allowed the initial infection to become established, which allowed these mice to develop immunity. However, for the animals treated with 50 mg/kg/day, the infection did not become established because the higher dose of SQ109 eliminated the parasites before the mice developed a robust immune response. Hence, these animals all died with the challenge at 42 days, a not uncommon observation in the development of drugs to treat *Toxoplasma* infections [23,24].

### 3.6. Activity of SQ109 Metabolites in Protozoa

Since 16 likely metabolites of SQ109 were reported active against *M. tuberculosis*, we assessed their activity in the in vitro model of *Leishmania* and *Trypanosoma* [26]. For ease of comparison with the previous work, we shall use here the same numbering system as that reported earlier (Figure 7). Some representative dose-response curves are shown in Figure 8 and EC_50_ values for parasite growth inhibition, and CC_50_ values for host cell toxicity (all in μM units) are shown in Table 1. Dose-response data for all cells and all compounds are given in the Appendix A. As can be seen in Table 1, none of the known (or likely) SQ109 metabolites investigated previously have more potent activity than does SQ109 against the protozoa or human host cell line investigated. However, with the protozoa, we do find quite potent activity with some metabolites (Table 1). Moreover, we also find generally high correlations (Figure 9) between the activity of SQ109 in a given assay with the activity of each of the 6 mono-oxygenated SQ109 metabolites. For example, Figure 9A shows EC_50_ and CC_50_ data for SQ109 (y-axis) and compound **9** in *T. cruzi* epimastigotes (color coded in green), U2OS (red), *T. cruzi* amastigotes (blue), *T. brucei* BSF (orange), and *L. donovani* promastigotes (magenta). The R^2^ values for the results shown in Figure 3A–F are: 0.810 (**9**, Figure 9A); 0.985 (**12**; Figure 9B); 0.937 (**21**, Figure 9C); 0.857 (**22**; Figure 9D); 0.924 (**23**; Figure 9E); 0.962 (**24**; Figure 9F), on average then the R^2^ value is ~0.9. These correlations are quite high and indicate that these SQ109 metabolites penetrate both human and protozoal cells.

What is also of interest about our results is that in each system the decrease in metabolite activity from that seen with SQ109 is much less that seen in the mycobacteria where in *M. smegmatis* there was a 33× decrease, in *M. tuberculosis* a >25× decrease [26]. Here, the values are 5.3× (*T. cruzi* epimastigotes), 5.0× (*T. cruzi* amastigotes), 6.0× (U2OS), 7.9× (*L. donovani*) and 3.6× (*T. brucei*), on average a ~6× decrease.

## 4. Discussion

SQ109 is an anti-tubercular compound in clinical trials to evaluate efficacy in drug-resistant *M. tuberculosis*-infected patients and a convenient summary of the Russian Phase IIb work was recently reported [27]. As part of a drug repurposing strategy for kinetoplastid diseases, the activity of SQ109 has been evaluated against *T. brucei, T. cruzi* and *Leishmania* species in vitro by several research groups, and potent activity was reported against all three parasites [8,9,10,11]. However, there have been no reports evaluating the anti-parasitic efficacy of SQ109 in any animal models so in this study, we assessed the therapeutic potential of SQ109 in acute mouse models of *T. brucei* and *T. cruzi* infections, as well as *L. donovani*. We also tested SQ109 in vitro with the intracellular tachyzoite stage of *T. gondii*, in addition to examining its possible efficacy in a toxoplasmosis mouse model.

We first confirmed the in vitro activity of SQ109 against the parasite strains that were used in the in vivo models. For *T. brucei*, the IC_50_ value was 0.32 µM, which is comparable to a previous report of 0.078 µM [8]. For *T. cruzi*, the IC_50_ value was 12.6 µM in epimastigotes and 0.61 µM in intracellular amastigotes, similar to a previous report of 4.6 and 1.2 µM, respectively [11]. However, in trypomastigotes, the IC_50_ value was higher (0.65 µM) than in the previous report (50 nM). In the *Leishmania* testing, the IC_50_ value was 1.38 µM for the intracellular amastigote form of *L. donovani* and 0.175 µM against extracellular promastigotes. SQ109 was previously reported to have IC_50_ values of ~11 nM against the intracellular amastigote and ~500 nM for the promastigote form, of *L. mexicana* [9]. The reported inhibitory values for *L. donovani* are 7.2 nM and 630 nM for amastigotes and promastigotes, respectively [10]. The anti-amastigote activity in the current study was thus ~200 fold less. One possible reason for this difference in activity may be due to the use of different host cells. The previous report utilized J774 macrophages that originated from lymphoma cells from BALB/c mice, whereas phorbol 12-myristate 13-acetate (PMA)-differentiated THP-1 cells were used in the current study, and host cell-dependent in vitro activity of anti-leishmanial drugs in *L. donovani*, as well as in other species, is well documented [20,28]. In the case of *T. gondii*, the activity was evaluated for the first time and the IC_50_ for SQ109 in the intracellular tachyzoite assay was 1.82 µM, generally comparable to that found with the intracellular stages of *L. donovani* and *T. cruzi*. We next investigated the in vivo efficacy of SQ109 for each kinetoplastid parasite, as well as for *T. gondii*. The maximum tolerated dose (MTD) of the compound was reported [29] as 600 mg/kg, and the tested dose was selected based on the MTD in which we did not observe any adverse effects throughout the in vivo experiments. In the *T. brucei* acute model, SQ109 was not able to completely clear the parasites in the bloodstream and there was only a very modest effect on survival. In contrast, there was an 80% survival rate observed with SQ109-treated *T. cruzi* infected mice, versus 25% with the control. This is better than the previous observation with amiodarone and warrants future combination-therapy studies with azoles such as posaconazole and itraconazole. In a previous pharmacokinetics study of SQ109, the reported C_max_ was 0.409 µM (135 ng/mL) and the t_1/2_ (terminal phase) was 5.2 h with a single 25 mg/kg dose *per os* administration [30]. The in vitro IC_50_ values for SQ109 against *T. brucei* (0.32 µM) and *T. cruzi* (12.6 µM) with 3 days of compound exposure are similar to the reported C_max_ value. However, the t_1/2_ of SQ109 is relatively short compared to the compound incubation time in vitro, and the IC_100_ values are expected to be higher than the C_max_. Additionally, the reported plasma protein binding of SQ109 is 5.8–8.5% in mice resulting in ~ 90% of SQ109 in mice plasma being expected to be in the free-drug fraction [31]. We should also note here that SQ109 is known to be rapidly metabolized in liver [30,31] and in future work it may be desirable to develop SQ109 analogs that are more metabolically stable.

In contrast with the acute models for *T. brucei* and *T. cruzi*, the organs in which the parasites are detected for the VL mouse model are liver and spleen. In particular, the LDU in liver is routinely determined for assessing the activity of anti-leishmania drugs. In a consecutive 28-day oral dosing period using 10 mg/kg, the tissue accumulation trend was reported to be similar to single dosing and an SQ109 level of ~2000 ng/g was maintained in the liver during the administration period [30]. Assuming the same volume-to-mass ratio, the SQ109 liver concentration is ~6 µM, which is higher than the in vitro IC_50_ value (1.38 µM). However, in a single dose (25 mg/kg, per os) tissue distribution study, the SQ109 concentration rapidly decreased at 4 and 10 h after the administration. This rapid decrease may have played a role in the lack of efficacy in the current VL acute model [30]. In a rat model, the SQ109 level was significantly higher in liver compared with other organs, which may imply that efficacy could differ when different VL experimental models, such as the hamster chronic model, are considered [31].

The experiment for *T. gondii* was performed with a hypervirulent RH strain in which the non-treated mice all died within 10 days post-infection. Eighty percent of mice treated with SQ109 at 50 mg/kg were protected against the initial infection. This result may be because the infection and compound treatment were each delivered via the intraperitoneal (i.p.) route, with only a 6-h time difference, and also because of the prolonged exposure throughout the 10 days of consecutive treatments. When mice were challenged again 42 days post initial infection, the 20-mg/kg-dose group showed a higher rate of protection, whereas the 50-mg/kg group all died, suggesting that rapid parasite clearance may not have allowed enough time to for the higher dose group to develop immunity. The in vitro activities of SQ109 against the kinetoplastid parasites and *T. gondii* were similar to that found against *M. tuberculosis*, however, the efficacy in the parasite in vivo models was less. One likely reason is the differences in the terminal target organs for the drug, and for the pathogens. In *M. tuberculosis*, the lung is both the primary organ of infection and the tissue with the highest concentration of SQ109, which is not the case with the kinetoplastids or *T. gondii*.

In animals, SQ109 is rapidly metabolized by cytochrome P450 enzymes [31], plus, it is metabolized inside *M. tuberculosis* [32] to a geranyl chain-terminal-oxidized species. In *M. tuberculosis,* it has been suggested that SQ109 metabolism might be important for activity [32,33]. In this study we tested 16 metabolites of SQ109 and 6 mono-oxygenated forms were found active where each of the metabolites showed good correlation with SQ109 across the tested protozoans. However, the metabolites were not as potent as SQ109 and this is consistent with the results obtained previously with *M. tuberculosis*, as well as with *M. smegmatis, Bacillus subtilis* and *E. coli* [26]. The lack of activity against the non-tuberculosis bacteria is likely to be due to lack of an MmpL3 target. In the mycobacteria, SQ109 is thought to target primarily the trehalose monomycolate transporter MmpL3 in addition to acting as a protonophore uncoupler [12]. Although there is an MmpL3-like protein in *T. cruzi* [34], its function is not known, and MmpL3 proteins are not present in *T. brucei*, *L. donovani* or in humans. The most likely reason for the strong correlation between the activity of SQ109 and the activity of the mono-oxygenated metabolites is, therefore, likely to be due to a more “physical” effect, for example, differences in cell membrane solubility between the metabolites, leading to differences in uncoupling activity. However, as can be seen from the actual EC_50_ and CC_50_ results (for **9**, **12**, and **21**–**24**) shown in Table 1, the actual range in EC_50_ or CC_50_ for the 6 metabolites is rather small and would be hard to predict computationally, although activity is clearly decreased on addition of a single oxygen.

## 5. Conclusions

Overall, the in vivo results with SQ109 we have obtained with *T. cruzi* and *T. gondii* are encouraging, but those with *T. brucei* and *L. donovani* are not. With *T. cruzi*, the results obtained (80% survival) were better than those reported previously for amiodarone (40% survival), acting alone. This is of importance since a recent azole combination therapy study in dogs with amiodarone, which has similar effects on *T. cruzi* as does SQ109, was very promising. The next logical step will thus be to investigate SQ109 combinations with the azoles, in *T. cruzi*, which should be of particular interest given that SQ109 has activity in vivo and acts synergistically with the potent azole posaconazole, *in vitro*. Results with *T. gondii* are also encouraging since the RH strain used is hypervirulent, so having an 80% protection is significant. Given that there are no obvious common protein targets in the systems investigated and that SQ109 itself is a potent protonophore uncoupler, it appears likely that the active SQ109 metabolites also function as protonophore uncouplers but because of enhanced polarity, they are less effective, due to decreased lipid membrane binding. SQ109 analogs that are more hydrophobic may, therefore, have better antiparasitic activity since lipid membrane binding will be increased and work is in progress on such compounds.

## Figures and Tables

**Figure 1 biomedicines-10-00670-f001:**
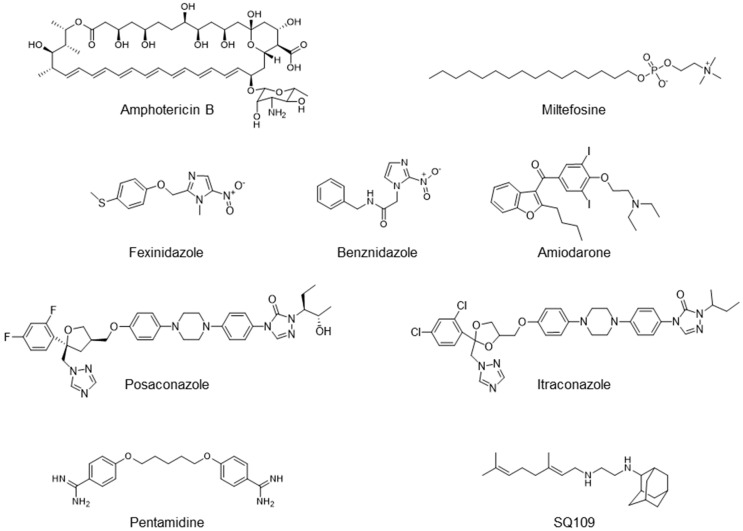
Structures of compounds described in this study.

**Figure 2 biomedicines-10-00670-f002:**
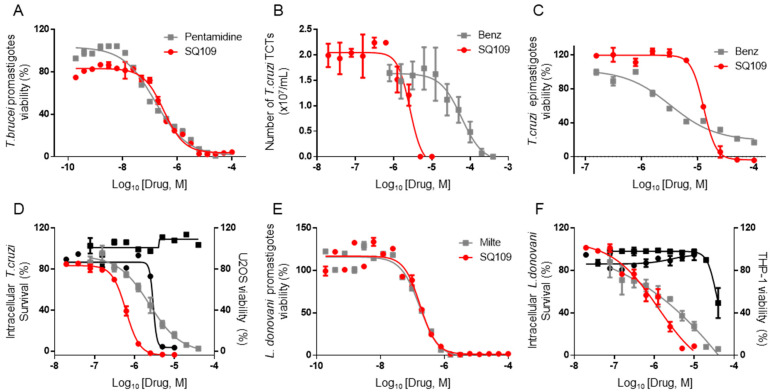
In vitro activity of SQ109 against kinetoplastid parasites. (**A**) Viability of *T. brucei* trypomastigotes treated with pentamidine [■] or SQ109 [●] for 72 h. (**B**) Number of *T. cruzi* TCTs (tissue culture trypomastigotes) treated with benznidazole [■] or SQ109 [●] for 24 h. (**C**) Viability of *T. cruzi* epimastigotes treated with benznidazole [■] or SQ109 [●] for 72 h. (**D**) Survival of intracellular *T. cruzi* amastigotes in U2OS treated with benznidazole [■] or SQ109 [●] and viability of U2OS treated with benznidazole [■] or SQ109 [●] for 72 h. (**E**) Viability of *L. donovani* promastigotes treated with miltefosine [■] or SQ109 [●] for 72 h. (**F**) Viability of intracellular *L. donovani* amastigotes in THP-1 followed by treatment of miltefosine [■] or SQ109 [●] and viability of THP-1 treated with miltefosine [■] or SQ109 [●] for 72 h.

**Figure 3 biomedicines-10-00670-f003:**
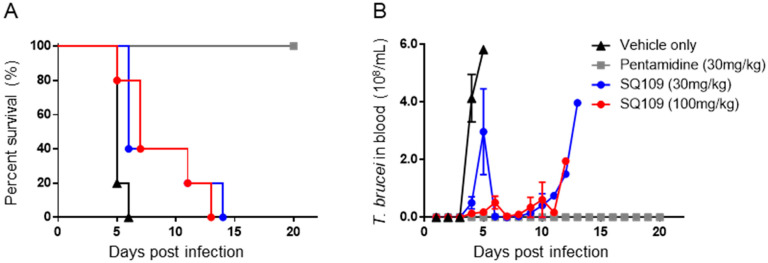
Efficacy of SQ109 in acute *T. brucei* mouse model. (**A**) Survival and (**B**) number of *T. brucei* parasites in the bloodstream of *T. brucei* infected BALB/c mice treated with vehicle only [▲], pentamidine (30 mg/kg) [■], SQ109 (30 mg/kg) [●] or SQ109 (100 mg/kg) [●]. (*n* = 5).

**Figure 4 biomedicines-10-00670-f004:**
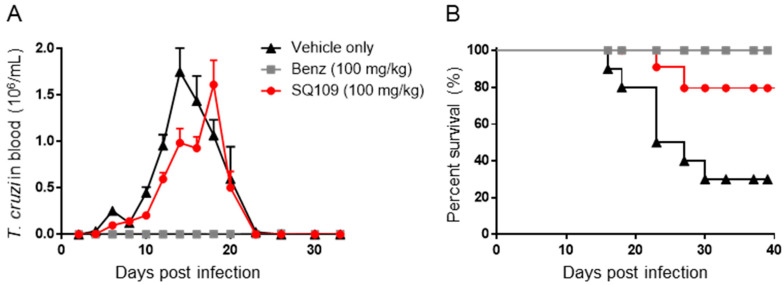
Efficacy of SQ109 in acute *T. cruzi* mouse model. (**A**) Number of *T. cruzi* parasites in the bloodstream. (**B**) Survival of *T. cruzi* infected BALB/c mice. In vivo study of BALB/c mice infected with *T. cruzi Dm28c*. of vehicle only [▲], Benznidazole (100 mg/kg) [■], or SQ109 (100 mg/kg) [●] treated BALB/c mice after infection of *T. cruzi Dm28c.* (*n* = 10).

**Figure 5 biomedicines-10-00670-f005:**
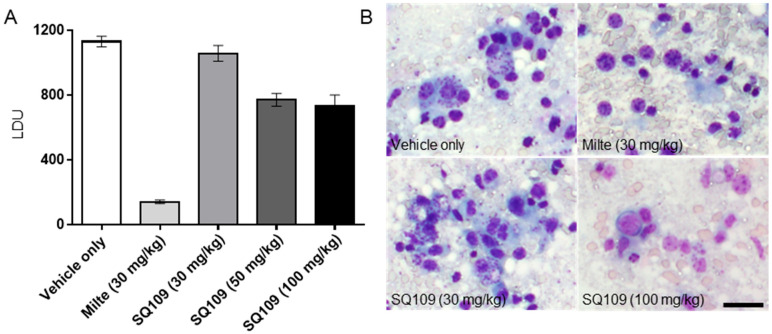
Efficacy of SQ109 in visceral *Leishmania donovani* mouse model. (**A**) Efficacy of tested drugs and controls in LDU (Leishman Donovan Unit). (**B**) Giemsa-stained images of BALB/c infected with *L. donovani* treated with vehicle only, miltefosine (30 mg/kg), or SQ109 (30, 50, or 100 mg/kg). (*n* = 5). Scale bar = 100 μm.

**Figure 6 biomedicines-10-00670-f006:**
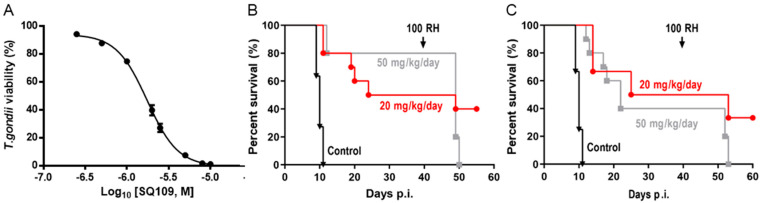
Effects of SQ109 on *T. gondii* cell growth inhibition in vitro and in vivo activity. (**A**) *T. gondii* cell growth inhibition. IC_50_ = 1.82 μM. *T. gondii* infected mice treated with vehicle only [▲], SQ109 (20 mg/kg) [●] or (50 mg/kg) [■] (**B**) by intraperitoneal route, or (**C**) by oral gavage (*n* = 10).

**Figure 7 biomedicines-10-00670-f007:**
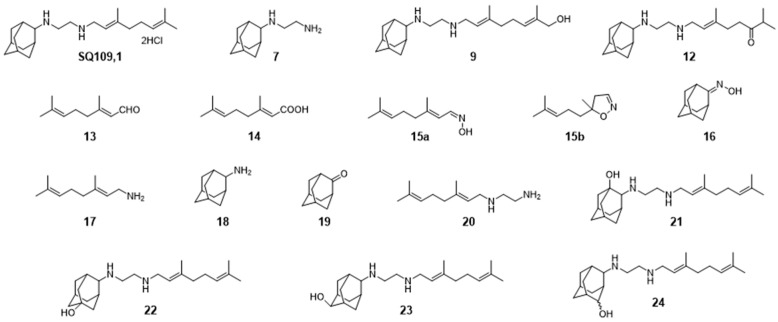
Structures of SQ109 and known/potential metabolites investigated. Note that, for clarity, compound numbers are those reported in previous work on *M. tuberculosis* [26].

**Figure 8 biomedicines-10-00670-f008:**
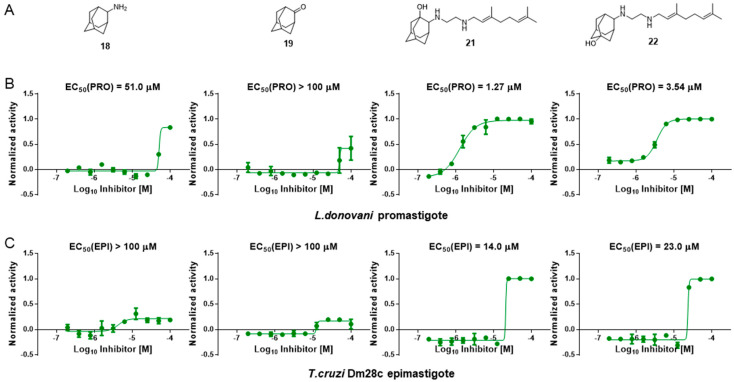
Representative dose-response curves, for *L. donovani* promastigote and *T. cruzi* epimastigote growth inhibition by compounds **18**, **19, 21** and **22**. (**A**) Structures of compounds investigated. (**B**) Dose-response curves for *L. donovani* promastigote growth inhibition. (**C**) Dose-response curves for *T. cruzi* Dm28c epimastigote growth inhibition.

**Figure 9 biomedicines-10-00670-f009:**
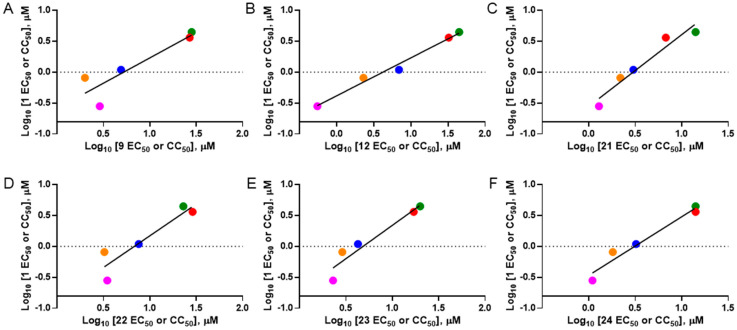
Correlations between metabolite activity with that of SQ109 in 4 assays: *T. cruzi amastigotes* (and the host U2OS cells), *T. cruzi* epimastigotes, *L. donovani* promastigotes and *T. brucei* BSFs. (**A**) Compound 9, r = 0.90 and *p* = 0.0373. (**B**) Compound 12, r = 0.99 and *p* = 0.0012. (**C**) Compound 21, r = 0.97 and *p* = 0.0062. (**D**) Compound 22, r = 0.93 and *p* = 0.0219. (**E**) Compound 23, r = 0.96 and *p* = 0.0095. (**F**) Compound 24, r = 0.98 and *p* = 0.0034. The color codes are *T. cruzi* epimastigotes = green (●), *T. cruzi* amastigotes = blue (●), U2OS = red (●), *T. brucei* BSF = orange (●), *L. donovani* promastigotes = magenta (●).

**Table 1 biomedicines-10-00670-t001:** Growth inhibition results for SQ109 and its known/potential metabolites against *T. cruzi* epimastigotes (EPI), *T. cruzi* amastigotes (AMA), *T. cruzi* amastigote host cells (U2OS), *L. donovani* promastigotes (PRO) and *T. brucei* bloodstream forms (BSF).

Compd.No.	Structure	*T. cruzi* Dm28c	*L. donovani*EC_50_ (PRO)μM	*T. brucei* BSFEC_50_ (μM)
EC_50_ (EPI)μM	EC_50_ (AMA)μM	CC_50_ (U2OS)μM
**1**	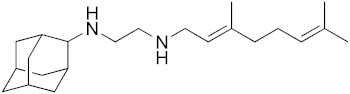	4.5	1.1	3.6	0.28	0.81
**7**	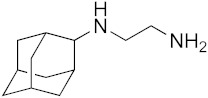	56	25	>100	22	4.6
**9**	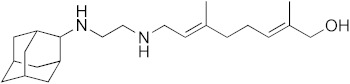	28	4.9	27	2.9	5.4
**12**	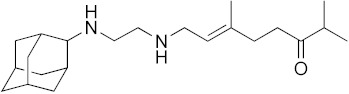	45	6.9	32	0.55	2.0
**13**	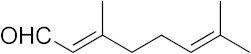	>100	>100	>100	>100	2.3
**14**	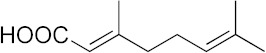	>100	>100	>100	>100	13
**15a**	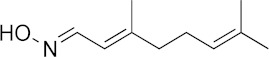	>100	>100	>100	>100	>200
**15b**	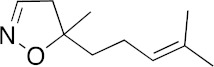	>100	>100	>100	>100	>200
**16**	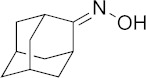	>100	>100	>100	>100	>200
**17**	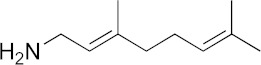	>100	40	>100	21	32
**18**	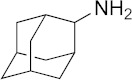	>100	50	>100	51	71
**19**	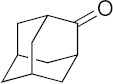	>100	>100	>100	>100	>200
**20**	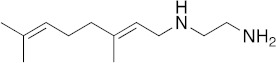	>100	>100	>100	6	2.2
**21**	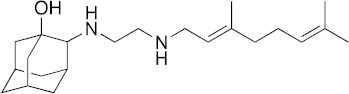	14	3	6.8	1.3	3.2
**22**	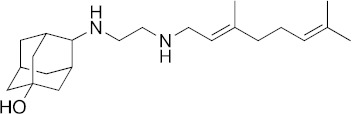	23	7.5	29	3.5	2.9
**23**	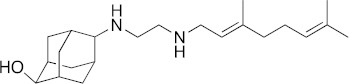	20	4.3	17	2.3	1.8
**24**	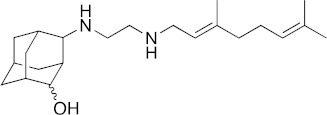	14	3.2	14	1.1	2.2

## Data Availability

Not applicable.

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
