# Peer review of "In Vivo Efficacy of SQ109 against Leishmania donovani, Trypanosoma spp. and Toxoplasma gondii and In Vitro Activity of SQ109 Metabolites"

_biomedicines, 2022, doi:10.3390/biomedicines10030670_

Round 1

Reviewer 1 Report

The manuscript presents an interesting study on the in vitro and in vivo activity of SQ109 against protozoan parasites causing neglected tropical diseases and other parasitic diseases. In addition, some of the SQ109 metabolites were also studied. Overall, the results obtained are promising, which could encourage the authors to continue these studies.

I really appreciate the experimental studies performed. The paper is well written and I am satisfied with the work.

Reviewer 2 Report

L20-25 The mouse in vivo drug efficiency results against Trypanosoma and Toxoplasma are described, but not against Leishmania.

L31 in vivo [italic]

L43 full parasite name should be [Trypanosoma brucei gambiense]

L84 when first mentioned full organism name should be used Leishmania donovani

Figure 3 and Figure 4. First figure should be described, cited in the text and afterwards should be the illustration

Figure 5 legend. Change leishmaniasis to Leishmania donovani

L303-304 delete sentence it repeats what was said in Introduction, also this sentence is not a part of results.

Figure 6 legend name is not correct. It should be in vitro and in vivo. L305 Figure 6A in vitro. L306 in vivo Figure 6B and 6C

Figure 6C the figure is cut, please place it correctly

L330 Figure 8 first mentioned and then L336 Figure 7 first mentioned, please correct

L332 some metabolites

L334 SQ109 metabolites

L328-338 it is description of previous work which is related this the present one. However, this section should not belong to results section of the present work. I suggest to move it to introduction.

Reviewer 3 Report

The topic of this manuscript is interesting and fits well the scope of the journal. The reviewer feels it can be accepted after some minor amendments.

1) This compound has been tested in Phase II/III trials. How good is its clinical safety?

2) What the half-life of such compound?

3) As the metabolites are active, is there any major species difference in metabolism between humans and mice?

4) For combination therapy, there is risk of drug drug interaction. What is the major  metabolic pathway of this drug?
